# Sertaconazole-Nitrate-Loaded Leciplex for Treating Keratomycosis: Optimization Using D-Optimal Design and In Vitro, Ex Vivo, and In Vivo Studies

**DOI:** 10.3390/pharmaceutics14102215

**Published:** 2022-10-18

**Authors:** Menna M. Abdellatif, Mina Josef, Mohamed A. El-Nabarawi, Mahmoud Teaima

**Affiliations:** 1Department of Industrial Pharmacy, College of Pharmaceutical Sciences and Drug Manufacturing, Misr University for Science and Technology, Giza 12566, Egypt; 2Department of Pharmaceutics and Industrial Pharmacy, Faculty of Pharmacy, Cairo University, El-Kasr El-Aini Street, Cairo 11562, Egypt

**Keywords:** keratomycosis, sertaconazole nitrate, D-optimal design, ocular drug delivery

## Abstract

This study aims to develop efficient topical therapy for keratomycosis using sertaconazolenitrate (STZN)-loaded leciplex (LP). The D-optimal design was used to optimize STZN-loaded LP by utilizing soy phosphatidylcholine (SPC) molar ratio (X_1_), cationic surfactant molar ratio (X_2_), and cationic surfactant type (X_3_) as the independent variables, whereas their impact was studied for entrapment efficiency percent (EE; Y_1_), particle size (PS; Y_2_), polydispersity index (PDI; Y_3_), zeta potential (ZP; Y_4_), and permeability coefficient (Kp; Y_5_). The optimized formula was evaluated regarding morphology, ex vivo permeation, mucoadhesion, stability, and in vivo studies. The optimized formula was spherical and showed EE of 84.87 ± 1.71%, PS of 39.70 ± 1.35 nm, PDI of 0.242 ± 0.006, ZP of +54.60 ± 0.24 mV, and Kp of 0.0577 ± 0.0001 cm/h. The ex vivo permeation study revealed that the optimized formula enhanced the Kp and corneal deposition by 2.78 and 12.49 folds, respectively, compared to the aqueous drug dispersion. Furthermore, the optimized formula was stable and revealed promising mucoadhesion properties. Finally, the in vivo studies showed that the optimized formula was superior to the drug dispersion in treating rats with induced keratomycosis. These results confirmed the capabilities of LP as a promising nanocarrier for treating ocular diseases topically.

## 1. Introduction

Keratomycosis, or fungal keratitis, is a potentially serious consequence that can result in visual abnormalities or even blindness if treatment is delayed [1]. Most keratomycosis infections are caused by *Fusarium* spp., *Candida* spp., and *Aspergillus* spp. [2]. Topical eye drops are the most convenient dosage form for treating keratomycosis due to ease of application and high patient compliance compared to other ocular dosage forms. However, the complex anatomy of the eye represents a challenge in achieving adequate ocular drug concentration, thus negatively affecting the clearance of the fungal infection [3]; therefore, poor ocular bioavailability, limited retention time, and low ocular tissue penetration are areas of chief concern in the topical use of antifungal agents [4]. Additionally, most antifungal drugs possess a limited aqueous solubility, which is still a challenging issue for developing a suitable formulation; many antifungal agents are used off-label as ocular eye drops and ointment [5].

Therefore, developing an efficient ocular drug delivery system capable of extending ocular drug retention and improving the corneal penetration and absorption of antifungal drugs through ocular barriers is of great importance.

Nanotechnology offers several significant benefits, including long-term medication release and precise tissue targeting. As a result, clinical pharmacologists have spent the last decade working to produce nanomedicine that can overcome obstacles to give a prolonged and tailored release with low toxicity [6]. Several research works have studied the implementation of nanoparticles as a successful ocular drug delivery system for antifungal drugs, such as polymeric nanoparticles, solid lipid nanoparticles, vesicular systems, and nanostructured lipid carriers. Incorporating an ophthalmic drug in a nanostructure improves the solubility, permeability, stability, and ocular bioavailability of the drug [7].

Sertaconazole nitrate (STZN) is an imidazole antifungal drug that is very effective in treating Candida albicans infections and some species of Fusarium and Aspergillus [8]. However, it has poor aqueous solubility, limiting its formulation to a suitable ocular delivery system. Several studies were performed to enhance STZN ocular delivery; Younes et al. developed STZN-loaded cubosomes, and the ex vivo corneal permeation showed that the cubosomal formula significantly enhanced the transcorneal permeability of STZN [9]. The same workgroup also studied the impact of loading STZN into binary mixed micelles, and the optimized mixed micelles formula succeeded in enhancing solubility of STZN up to 338.82-fold and consequently enhanced corneal permeation and retention [10]. Additionally, Tavakoli et al. formulated in situ thermosensitive gel containing STZN-loaded nanostructured lipid carriers for prolonged ocular drug delivery. The optimized gel formula exhibited a significantly higher antifungal activity than free STZN [11]. The previous studies pointed out that several nanocarrier systems had been modified to enhance STZN corneal penetration; however, there is a need to develop a scalable, biocompatible, and simple-to-prepare drug delivery system that can provide a significant enhancement in STZN corneal permeation and retention.

Leciplex (LP) is a self-assembled, lecithin-based cationic nanocarrier; the main ingredients of the LP system are phospholipid; a cationic surfactant (SAA); and a biocompatible surfactant, such as Transcutol HP. LP provides several advantages over other nanocarrier systems, such as ease of preparation (the system is prepared in one fabrication step), the absence of organic solvent during its formulation, and the simplicity of scale-up [12]. 

In this study, LP was chosen as a potential nanocarrier for enhancing the corneal penetration and deposition of STZN as the presence of a positive charge facilitates vesicle adhesion onto the negatively charged sialic acid moieties found in the mucus membrane on the corneal surface, which results in higher corneal retention, minimizing drug elimination via lachrymal flow and improving transcorneal flux. Additionally, the apparent advantage of small particle size is that it creates a large contact surface with the ocular surface, allowing enhanced corneal absorption [13].

Therefore, STZN-loaded LP formulae were formulated via a single-step fabrication method. A D-optimal design was utilized to study the influence of different variables on the characteristic of LP and compare their influence. The preliminary studies found several independent factors, such as lipid-to-cationic-surfactant molar ratio and surfactant type, to impact the LP characteristics. Therefore, the phospholipid molar ratio (X_1_), cationic SAA molar ratio (X_2_), and cationic SAA type (X_3_) were chosen as independent factors for the design, whereas entrapment efficiency (EE%; Y_1_), particle size (PS; Y_2_) polydispersity index (PDI; Y_3_), zeta potential (ZP; Y_4_) and in vitro drug release in terms of permeability coefficient (Kp; Y_5_) were chosen as dependent factors. The optimized formula was selected using Design Expert^®^ 7 software and evaluated in terms of morphology, ex vivo permeation, mucoadhesion, differential scanning calorimetry (DSC), stability study, and in vivo study in which histopathological study and assessment of the inflammatory biomarkers were used to evaluate the efficacy of STZN loaded LP.

## 2. Materials and Methods

### 2.1. Materials

Sertaconazole nitrate (STZN) was kindly gifted by October Pharma (Cairo, Egypt); Transcutol^®^ HP was obtained from Gattefosse India Ltd. (Mumbai, India); and soy phosphatidylcholine (SPC), cetyltrimethylammonium bromide (CTAB), and dimethyldidodecylammonium bromide (DDAB) were purchased from Sigma-Aldrich (St. Louis, MO, USA) All other reagents and solvents were of HPLC analytical grade and were obtained from Fisher Scientific Company (Waltham, MA, USA). 

Wistar albino rats without ocular damage or diseases were obtained from Misr University’s for Science and Technology animal center (Giza, Egypt). The ethical committee approved all animal studies of the Cairo University faculty of pharmacy.

### 2.2. Preparation of SZTN-Loaded LP

LP formulae were prepared using a simple one-step method with varied lipid: SAA molar ratios (Table 1). SPC, cationic SAA, and drug (50 mg) were first solubilized in 0.5 mL Transcutol HP at 70 °C; 9.5 mL distilled water was heated at the same temperature and then added to the lipid mixture with continuous mixing using a digital hot plate magnetic stirrer (LX653DMS, LabDex, London, UK) at 900 rpm for 15 min. A homogeneous nano-dispersion was produced and stored in the refrigerator at 4 °C until evaluation [14].

### 2.3. Characterization and Optimization of STZN-Loaded LP

#### 2.3.1. Determination of Entrapment Efficiency (EE%)

One mL of STZN-loaded LPs was filled in the dialysis bag (MW of 12,000–14,000 Dalton) and suspended in 50 mL distilled water (containing 25% methanol) while magnetically agitated at 50 rpm. Aliquots were removed at specific time points, and STZN concentrations were measured at 261 nm using a UV spectrophotometer (UV-1650; Shimadzu Corp., Kyoto, Japan) until a constant concentration was obtained. The following equation was used to calculate the EE percent [15]:(1)$$\mathrm{EE}\;\left(\%\right)=\frac{\mathrm{Total}\;\mathrm{STZN}\;\mathrm{amount}\;-\mathrm{Diffused}\;\mathrm{STZN}\;\mathrm{amount}}{\mathrm{Total}\;\mathrm{STZN}\;\mathrm{amount}}\times 100$$

Each evaluation is the average of three individual experiments ± SD. 

#### 2.3.2. Determination of Particle Size (PS), Polydispersity Index (PDI), and Zeta Potential (ZP)

Malvern Zetasizer (Malvern Instruments Ltd., Malvern, UK) measured the PS, PDI, and ZP of the LP formulae using the light scattering method at 25 °C. Following proper dilution, the assessments were carried out [16].

#### 2.3.3. In-Vitro Drug Release Study

Using a locally made Franz’s diffusion cell with an effective release area and receptor cell volume of 3.14 cm^2^ and 50 mL, respectively, the in-vitro drug release of STZN from LP formulae was studied. The temperature of the receiver vehicle (PBS, pH 7.4 containing 25% methanol) was kept at 37 ± 1 °C and was constantly stirred by a magnetic stirrer at 50 rpm to ensure sink conditions. The cellulose membrane (cut off 12,000–14,000) was placed between the receptor and the donor compartments. The donor compartment was filled with 1 mL (equivalent to 5 mg of STZN) of LP formulae. Every hour, aliquots of 0.5 mL samples were taken and immediately subjected to UV spectrophotometer analysis. To maintain sink conditions, the same volume of fresh media was added to the receptor compartment to replace what had been taken out [17]. The amount of drug released was plotted versus time. The in vitro release parameters were calculated; the steady-state flux (J_ss_) was the slope of the linear portion (µg/cm^2^/h), C_0_ was the initial drug concentration (µg/cm^2^), and the cumulative percent of drug permeated after 8 h (Q_8)_ was also determined. The apparent corneal permeability coefficient (cm/h) was determined according to the equation Papp = J_ss_/C_0_. The in vitro release profiles were fitted to various kinetic models (Higuchi, first-order, Peppas, and zero-order) to determine the drug release mechanism.

### 2.4. Optimization of STZN-Loaded LP Formulations Using D-Optimal Designs

Design Expert^®^ software version 13 (Stat Ease, Inc., Minneapolis, MN, USA) was used to perform a D-optimal design experiment to study the influence of several parameters on the fabrication of LPs. The three tested parameters were (X_1_: lipid molar ratio), (X_2_: cationic SAA molar ratio), and (X_3_: cationic SAA type) at two levels in the design of LP formulation. In both designs, the EE% (Y_1_), PS (Y_2_), PDI (Y_3_), ZP (Y_4_), and Kp (Y_5_) were chosen as dependent variables, as shown in Table 1.

#### Selecting the Optimized STZN-LP Formula

The optimal formula was chosen using the desirability tool. The base for selecting the optimized formula was to produce the least PS and PDI and the highest Kp, EE%, and ZP values.

### 2.5. Morphology of the Optimized Formula

The morphology of the optimized formula was assessed using a transmission electron microscope (TEM) (JEM-1230, Joel, Tokyo, Japan). The LP formula was diluted 10-fold, placed on the carbon-coated grid, then negatively stained with 1% phosphotungstic acid before being dried for visualization [18].

### 2.6. Ex-Vivo Corneal Permeation Studies

Ex vivo corneal permeation experiments were carried out using a modified Franz diffusion cell with a diffusion area of 0.785 cm^2^. The fresh cow cornea was fixed between the donor and receptor chambers. An amount of 1 mL of LP dispersion, equivalent to 5 mg of STZN, was accurately measured and placed in the donor cells. The receptor compartment was filled with 25 mL of phosphate buffer saline solution (pH 7.4) containing 25% methanol to ensure sink condition and maintained at 37 ± 1 °C under magnetic stirring at 50 rpm. Every hour, 0.5 mL of permeation media was withdrawn, and an equal volume of fresh media was added to the receiver cell. The samples were filtered through a 0.45 μm membrane and analyzed using a validated HPLC method, for which the mobile phase was composed of acetonitrile disodium phosphate buffer (pH 4) (40:60, *v*/*v*). The drug was separated into a Luna C18 column using a flow rate of 1.8 mL/min. The retention time of the drug was around 21 min [19]. The experiments were performed in triplicate. The amount of drug permeating through the corneal epithelium was plotted versus time. The apparent corneal permeability coefficient (cm/h) was determined according to the equation Kp = J_ss_/C_0_, where J_ss_ (steady stat flux) was the slope of the linear portion (μg/h·cm^2^) and C_0_ was the initial drug concentration (μg/cm^2^) [20]. 

After the completion of the ex vivo permeation study, the cornea was washed 3 times with distilled water to completely remove the drug then soaked in 20 mL methanol overnight and sonicated for 15 min in a bath sonicator (Elmasonic S 60 H, Elma, Bangkok, Thailand). Finally, aliquots were withdrawn, filtered through a 0.45 μm membrane, and analyzed using a validated HPLC method as mentioned before to detect the amount of drug deposited in the corneal tissue after 8 h [21].

### 2.7. Confocal Laser Scanning Microscopy Study (CLSM)

The drug in the optimized LP formula was replaced by 1% *w*/*v* fluorescein diacetate (FDA). Cow cornea was fixed in diffusion chambers with the previous aspects of the ex-vivo permeation study. FDA-loaded LP was applied to the corneal surface and remained there for 8 h. Longitudinal sections were placed in paraffin wax, cut into sections using a microtome (Rotary Leica RM2245; Leica Biosystems, Wetzlar, Germany), and were examined for fluorescence in the corneal tissues. The slides were visualized using an inverted microscope (LSM 710; Carl Zeiss, Oberkochen, Germany). The excitation and emission wavelengths of the FDA were λ_max_ 497 nm and λ_max_ 516 nm, respectively. The cornea was optically scanned under a 40× objective lens (EC-Plan Neofluar 63×/01.40 Oil DICM27). Confocal images were supplied by LSM Image Browser software, release 4.2 (Carl Zeiss Microimaging GmbH, Jena, Germany). Light intensity was also measured [22]. 

### 2.8. Differential Scanning Calorimetry

To demonstrate the thermal characteristics of CTAB, STZN, SPC, and the optimized LP formula, differential scanning calorimetry (DSC) was used. A thermal analyzer was used to record the DSC thermograms (TA-60, Shimadzu, Japan). The samples were heated from 25 to 300 °C at a continuous rate of 10 °C/min while hermetically sealed in aluminum pans [23]. 

### 2.9. Effect of Storage on the Optimized LP Formula

The capability of LP to preserve its size and charge during the storage period was examined. The optimized formula was kept at 4 °C temperature for 3 months and then examined by comparing the PS, PDI, and ZP of the stored formula with the fresh formula. Additionally, the system was visually observed for sedimentation [24]. 

### 2.10. Evaluation of Mucoadhesion Aspects of the Optimized Formula

The mucoadhesive properties of the optimized formula were investigated by mixing the same volume of the LP with a 1% *w*/*v* aqueous solution of porcine mucin. The mixture was mixed for five minutes, then left to reach equilibrium overnight. Zetasizer was used to estimate the charge of mucin and the charge of mucoadhesive nanovesicles in the presence of mucin [25]. 

### 2.11. In-Vivo Assessment of the Optimized LP Formula

All the animal care and experiments in this assessment were authorized via the Research Ethics Committee of the Faculty of Pharmacy, Cairo University (Approval No. PI 2845), adhering to the “Guide for the Care and Use of Laboratory Animals” declared via the Institute of Laboratory Animal Research (Washington, DC, USA).

Twenty-four Wistar albino rats (weighing 150 g) were obtained from the Misr University for Science and Technology animal center. All the guiding principles in the care and use of laboratory animals are strictly adhered to throughout the study. The rats were to be kept in an environment of controlled temperature (24–26 °C) and controlled photoperiod (12 h of light and 12 h of dark) for one week before starting the experiment. They were housed individually in stainless steel cages and given ad libitum access to food and water. Eighteen rats received daily immunosuppressant therapy in the form of dexamethasone 1 mg/L (Merck, Darmstadt, Germany for 10 days, while the other six rats were set as the negative control (GPI) [19]. Two days before the experiment, Levofloxacin eye drops (Santan, Ishikawa, Japan) were administered daily. The immunosuppressant rats were deprived of food but had free access to water 24 h before the experiment. All the immunosuppressant rats were sedated using an intramuscular injection of ketamine (30 mg/kg) [26]. The right corneas of all rats were marked using 7 mm corneal trephine, followed by the scrapping of corneal epithelium. Then corneas were inoculated with candida using a needle of 27 gauge to inject 0.1 mL of Candida species with a concentration of 5 × 10^5^ cells/mL into the posterior corneal stroma [27]. In either case, the left eye was kept as a control and received an equal volume of saline solution. The procedure was performed under complete aseptic precautions with a binocular microscope, and the eyes were washed with chloramphenicol eye drops. Forty-eight hours later, all corneas of the immunosuppressant rats were examined for signs of infection to confirm the flourishing of the organism [28]. The infected rats were divided into three groups, each of 6, and the first group was set as the positive control (GP II) and received no treatment. The second group received the optimized 50 µL of LP formula three times daily for 14 days (GP III). The third group received the equivalent dose using the aqueous drug suspension as the comparable dosage forms (GP IV). The negative control (GP I) received 50 µL of normal saline instead. 

#### 2.11.1. Histopathology Study

At the end of the therapy, three rats from each group (*n* = 3) were utilized for the histopathology examination. First, the whole eyes were kept in formalin saline solution (10% *v*/*v*). Secondly, the eyes were dissected, and the ocular samples were dehydrated with alcohol and fixed in melted paraffin. Then, a microtome was used to arrange thin sections (2 mm), deparaffinized, and stained with hematoxylin and eosin. Finally, a light microscope (DMS1000 B; Leica, Milton Keynes, UK) was used to examine the samples [29].

#### 2.11.2. Assessment of Serum Inflammatory Biomarkers

Three rats from each group were given anesthesia, and blood samples were collected into tubes via retro-orbital puncture for assessment of high-sensitivity C-reactive protein (hs-CRP), interleukin-23, and beta-D-glucan (BDG) using enzyme-linked immunosorbent assay (ELISA) according to the manufacturer’s instructions. The assay process involved measuring the absorption of the yellow color at 450 nm. The sample concentration was determined according to the standard curve [19]. 

### 2.12. Statistical Analysis of Data 

To investigate the significant difference between the results of studied formulae, the one-way analysis of variance (ANOVA) test was used. The significance level was set at 0.05, and (* *p* < 0.05) was considered statistically significant.

## 3. Results and Discussions

### 3.1. Factorial Design Optimization

Preliminary trials were performed to detect the independent variables’ promising ranges. The independent variables that were studied for LPs are (X_1_: lipid molar ratio), (X_2_: cationic SAA molar ratio), and (X_3_: type of the cationic SAA). Nineteen LP formulae were formulated and characterized to study the impact of these independent variables on EE% (Y_1_), PS (Y_2_), PDI (Y_3_), ZP (Y_4_), and Kp (X_5_), as seen in Table 2. Note the design analysis values in Table 3.

### 3.2. Effect of Formulation Variables on the EE%

The significance of the independent variables, X_1,_ X_2,_ and X_3_, on the EE% of STZN in the LP formulae is illustrated in Table 3 and Figure 1a,b. The EE% of STZN LP formulae ranged from 68.50 ± 1.2 to 96.65 ± 0.80%, as shown in Table 2. The results showed that lipid molar ratio (X_1_) significantly increased the EE% (*p* < 0.0001). This result might be attributed to the lipophilic nature of STZN (log *p*, 6.23), as higher lipid concentrations result in higher amounts of STZN solubilized in the lipid portion; thus, an increase in entrapment efficiency was observed. On the contrary, increasing the cationic SAA molar ratio significantly reduced EE% (*p* ≤ 0.0001). These results may be due to the solubilization of phospholipid by the cationic SAA, leading to decreasing STZN encapsulation into the LP. Contrary to previous literature that pointed out the positive impact of the double-chain cationic SAA (DDAB) with a higher log *p*-value (11.8) on the EE% of lipophilic drugs compared to the single chain CTAB with a lower log *p*-value 8 [30,31], CTAB increased the EE% of STZN compared to DDAB (*p* = 0.0084). DDAB is a double-chain SAA that could compete with the positively charged drug STZN for affinity with the negative phospholipid groups; therefore, by increasing the DDAB molar ratio, the amount of STZN entrapped was reduced. These results agree with Esposito et al., who found that the association of the negatively charged desoxiribonucleotide defibrotide (DFT) increased in the presence of DDAB more than CTAB, as the number of positive charges able to bind to the negatively charged DFT was higher than CTAB [32]. The regression equation of the fitted model was EE = +88.84 + 4.59 × A − 4.38 × B − 1.87 × C + 1.44 × A × B + 0.95 × A × C − 2.38 × B × C.

### 3.3. Effect of Formulation Variables on the PS

The optimal PS to avoid local ocular irritation and ensure high absorption by the corneal or conjunctival routes is less than 800 nm [33]. The PS of LPs formulae ranged from 24.18 ± 0.11 to 106.84 ± 0.19 nm, as displayed in Table 2 and Figure 2a,b. ANOVA results indicated that X_1_, X_2_, and X_3_ significantly influenced the PS of the prepared LP. The lipid molar ratio (X_1_) directly impacted the PS (*p* < 0.0001), and increasing lipid concentration resulted in a larger PS. These findings agree with EE% outcomes, in which increasing lipid concentration resulted in a higher entrapment of STZN in the prepared LP, which led to increased PS. Additionally, at a high SPC molar ratio, the amount of SAA will not be enough to reduce the interfacial tension, leading to the formation of larger LPs. For the cationic SAA molar ratio (X_2_), smaller vesicles were obtained at a high cationic SAA molar ratio due to lipid solubilization with SAA. The SAA type significantly influenced the PS of the prepared LP, as CTAB produced larger vesicles (*p* < 0.0001). These results agree with Hassan et al. and Varghese et al. They found that the inclusion of the double-chain SAA (DDAB) led to the formation of smaller vesicles due to its surface stabilization property [34,35]. Additionally, the increase in drug encapsulation is usually accompanied by an increase in particle size [36]. The regression equation of the fitted model was PS = +53.78 + 18.59 × A − 16.3 × B − 3.30 × C − 2.67 × A × B − 1.25 × A × C − 2.77 × B × C + 6.03 × A^2^ + 8.05 × B^2^.

### 3.4. Effect of Formulation Variables on the PDI

The PDI values for all LP formulae were lower than 0.3, indicating the homogeneity of the formulae [37]. Upon analysis of the independent variables’ effect on PDI (Y_3_), there was no significant model fit to the data. 

### 3.5. Effect of Formulation Variables on the ZP

The ZP values of the LP formulas ranged from (−39.20 ± 0.41 to +54.60 ± 0.24 mV), as displayed in Table 2 and Figure 3. ANOVA results showed that X_1_ and X_2_ significantly influenced the ZP of the prepared LP with *p* values of 0.0035 and <0.0001, respectively. The lipid molar ratio (X_1_) showed a significant impact on the ZP values where the LP formula (F1) is composed of 3:0 (SPC: cationic SAA) on increasing the lipid molar ratio to 5:0 (F18), the ZP value increased from −32.5 ± 0.43 to −39.20 ± 0.41 mV. It was expected that by increasing the SPC molar ratio, the negativity of the ZP values would increase, as SPC bears a net negative charge on its surface in a medium of low ionic strength in which the polar head group is directed so that the negatively charged phosphatidyl group is oriented to the outside and the positively charged choline group is placed on the inside. The addition of the cationic SAA affected the ZP values of the prepared LP formula as it shifted the ZP values from negative values to positive values, and by increasing the SAA molar ratio (X_2_), the ZP values increased, leading to the formation of a more stable nanovesicles. The positive charge could allow for a prolonged drug contact time through ionic interaction with the negatively charged sialic acid residues present in the mucous of the cornea and conjunctiva [38]. Finally, the cationic SAA type (X_3_) showed a non-significant difference with a *p*-value of 0.7253. The regression equation of the fitted model was ZP = −4.60 − 15.29 × A + 28.35 ×B − 1.40 × C.

### 3.6. Effect of Formulation Variables on the In Vitro Drug Release

The in vitro drug release from different LP formulae was represented in terms of the permeability coefficient (Kp). The values of Kp of STZN from LP formulae ranged from 0.01989 ± 0.0001 to 0.05770 ± 0.0001 cm/h, as displayed in Table 2 and Figure 4. The cumulative amount of STZN released after 8 h (Q8) ranged from 209.17 ± 2.09 to 1173 ± 2.56µg/cm^2^, while J_ss_ ranged from 36.43 ± 0.99 to 91.89 ± 0.43 µg/cm^2^/h. ANOVA results showed that X_1_ and X_2_ significantly influenced the release of the prepared LP. Regarding lipid molar ratio (X_1_) had a direct negative relation with (*p* < 0.0001), and drug permeation from vesicles decreased as the lipid concentration increased due to an increase in the rigidity of the nanoparticles. For (X_2_) cationic SAA, smaller vesicles were formed at a high cationic SAA molar ratio; therefore, a higher surface area was also obtained. Furthermore, lipid solubilization with SAA led to higher values of Kp from LPs. The cationic SAA type (X_3_) showed a non-significant difference with a *p*-value of 0.0899. The regression equation of the fitted model was Kp = +0.029 − 8.557 × 10^−3^ × A + 8.240 × 10^−3^ × B − 1.328 × 10^−3^ × C − 5.890 × 10^−3^ × A × B + 1.202 × 10^−3^ × A × C − 1.778 × 10^−3^ × B × C − 5.783 × 10^−3^ × A^2^ + 7.625 × 10^−3^ × B^2^.

Figure 5 shows the in vitro release profiles of LP formulae (F4, F1, and F15) with lipid-to-SAA molar ratios or 5:0, 3:0, and 1:0, respectively (LP formulae composed of SPC only) and their comparable formulae (F3, F5, and F2) with lipid-to-SAA molar ratios 5:1, 3:1, and 1:1, respectively. The in vitro release profiles in all formulae composed of SPC alone were characterized by an initial burst release followed by a sustained release. Burst release occurred due to the presence of some of the STZN on the external surface of the LP. The lipophilic nature of the STZN could be the reason for the sustained release of the drug from the internal lipid phase after the initial burst release. On the contrary, for the formulae composed of SPC and SAA, the in vitro release profiles are characterized by an increased rate of initial burst release with diminishing the sustained release period, especially for the LP formula with a low lipid molar ratio (F2). This might be due to the lipid’s solubilization by SAA. LP formula F2 (composed of 1:1 SPC to DDAB) exhibited the highest cumulative amount of STZN released among the previously mentioned LP formulae; this enhancement might be due to the reduced PS (24.18 ± 0.11 nm); that created a high surface area or the low lipid molar ratio together with lipid solubilization effect of SAA decreasing the rigidity of the vesicle. The regression coefficient values indicated that the in vitro release profile of all LP formulae could best be fitted using the Higuchi release model.

### 3.7. Selection of the Optimized LP Formula

To select the optimized LP formula, specific parameters were set in Design Expert^®^ software version 13. These conditions chose nanovesicles with the highest EE%, ZP, Kp, and the lowest PS and PDI. The optimized formula that met these criteria was composed of CTAB as cationic SAA type and SPC with a molar ratio of 1:1. The desirability of the optimized formula was 0.759. Subsequently, this formula was selected as the optimized formula for further inspection.

### 3.8. Morphology

The TEM micrographs of the optimized formula showed that the LP was spherical, as shown in Figure 6. In addition, the particle size result determined by Zetasizer was in good agreement with TEM observations.

### 3.9. Ex Vivo Corneal Permeation Studies

The cornea is the primary pathway for intraocular penetration of topically applied medications [39]. From Figure 7 and Table 4, it could be concluded that the quantity of STZN permeated through the cornea and deposited in the corneal tissues from LP was significantly (*p* < 0.05) greater than that from STZN aqueous dispersion. In addition, the permeability coefficient (Kp) was increased 2.78-fold and the corneal deposition 12.49-fold. The remarkable enhancement in the corneal deposition compared to corneal permeation referred to the capability of the LP to create a reservoir or depot that releases the drug slowly after endocytosis by the epithelial cells of the cornea. This enhancement in the corneal permeation and deposition might be attributed to improved corneal retention time due to electrostatic attraction between the positively charged LP and the negatively charged mucus. Additionally, vesicle size affects drug permeation. The smaller the vesicle size, the greater the interfacial area available for drug exchange through the hydrated network of the corneal stroma, consequently improving the clinical efficacy of the drug. These results agree with Wang et al., who correlated the improvement of the corneal permeation of puerarin and scutellarin with the presence of the positive charge, which prolonged the residence time of the nanoparticle on the ocular surface and led to the formation of a drug depot [40].

### 3.10. Confocal Laser Scanning Microscopy Study (CLSM)

The CLSM results were correlated with the results of the ex-vivo corneal permeation studies, where Figure 8 shows that the FDA-loaded LP exhibited deposition of fluorescence in all the corneal layers. Scans were taken from the longitudinal section that supplied information about the penetration depth of LP in corneal tissues. The light intensity was 12,012 ± 699.8.

### 3.11. Differential Scanning Calorimetry 

To further confirm the interaction between STZN and various LP components. DSC thermograms were recorded for STZN, SPC, CTAB, and the optimized LP formula, as shown in Figure 9. The STZN thermogram showed peaks at 160.11, and 206.08 °C with enthalpy values of −142.5 mJ and 515.04 mJ for each peak, respectively, and SPC showed broad peaks at −104.88 and −273.25 °C with enthalpy values of −307.22 mJ and −659.79 mJ for each peak, respectively. CTAB was at 102.4 °C and 261 °C with enthalpy values of −896.95 mJ and −2573.74 mJ for each peak, respectively. The thermogram of LP showed three peaks at 56.63, 142.41, and 254.34 °C with enthalpy values of −243.04 mJ, −206 mJ, and −238.07 mJ for each peak, respectively. The shifting of the SPC endotherm to lower temperature values might be due to the solubilization of SPC with CTAB. These results confirmed the interaction of CTAB with the phospholipid portion of the LP formula. Additionally, the transition enthalpy of the lyophilized formula was decreased due to the interaction of the LP’s various components, which perturbed the lipid bilayer’s packing [41]. Also, there was shifting in the endothermic peak of the STZN due to hydrophobic interactions between the drug and various LP components; these results agree with Salama et al., who found shifting in spironolactone endothermic peak upon loading into leciplex formulae [37].

### 3.12. Effect of Storage on the Optimized LP Formula

The effect of storage on the optimized formula was studied. The visual inspection of the formula did not demonstrate any sedimentation or vesicle aggregation during the storage period. In addition, EE%, PS, PDI, and ZP measurements were 83.22 ± 1.3%, 48.6 ± 1.9 nm, 0.22 ± 0.03, and 41.1 ± 1.4 mV, respectively. These results showed insignificant variation from the prepared LP (paired t-test, *p* > 0.05). These results assured the capability of the LP formula to maintain its characteristics during the storage period.

### 3.13. Evaluation of Mucoadhesion Aspects of Mucoadhesive Nanovesicles

The LP formula shifted the negative charge of mucin from −12.06 ± 1.99 mV to a positive surface charge of ZP value +16.40 ± 0.74 mV. In addition, a positive observation was gained from the complexes formed between mucin and LP, in which the negative charge of the mucin would be neutralized with the positive charge of the LP adhered on their surface. The results confirmed the mucoadhesive properties of the LP formula, as previous studies found that the component adhesion might alter the surface characteristics of the mucin if these components have a mucoadhesive property [42,43].

### 3.14. In Vivo Assessment of the Optimized LP Formula

A histopathological study was conducted to evaluate the efficacy of STZN-loaded LP compared to aqueous drug dispersion. Histological examination of H&E-stained sections of the cornea of the control albino rat (GP I) showed a normal histological structure of stratified squamous epithelium, stroma, and endothelium (Figure 10A). However, corneal sections obtained from untreated rats (GP II) revealed numerous structural alterations compared to control rats (GP I), including increased corneal thickness and decreased stratified squamous epithelium portions. In addition, some epithelial cells had vacuolated cytoplasm. Furthermore, the stroma had dispersed collagen fibers, proliferating inflammatory cells, edema, and dilated blood capillaries engorged with blood (Figure 10B).

On the contrary, corneal sections of rats treated with STZN-loaded LP appeared to have a normal histological structure of stratified squamous epithelium, stroma, and endothelium compared to rats with untreated fungal keratitis (Figure 10C). While corneal sections obtained from rats treated with aqueous drug dispersion (GPIV) revealed partial recovery evidenced by a decrease in the corneal thickness, an epithelium with restored architecture but still without normal layer arrangement, presence of few inflammatory cells, little collagen fiber dispersion, and blood capillaries appeared narrower with blood compared to untreated rats with fungal keratitis (Figure 10D). The results of the histopathological study were in good agreement with the results of the ex-vivo and CLSM results, as both studies pointed out the success of LP in enhancing corneal permeation and deposition of STZN compared to the drug dispersion. This enhancement might be due to the mucoadhesion property of LP that allowed attachment to the mucin layer that covered the conjunctiva and the cornea, ensuring intimate contact between the drug and the biological tissues, resulting in a high local concentration of the drug as well as its increased absorption through the biological membranes. Additionally, the small PS created a large contact surface with the ocular surface, enabling enhanced corneal absorption. Furthermore, the LP assured the sustained release of STZN and prolonged residence time in the corneal tissue. The enhancement in the STZN corneal permeation and deposition resulted in the superiority of STZN-loaded LP in treating keratomycosis compared to drug dispersion.

#### Assessment of Serum Inflammatory Biomarkers and BDG

Cytokines and other mediators play essential roles in eye inflammation. In this study, STZN-loaded LP restored the normal levels of the inflammatory mediators and BDG (*p* > 0.05) compared to the drug aqueous dispersions, as shown in Table 5. These results confirmed the outcomes of the histopathological study that revealed the enhancement of STZN-loaded LP in treating keratomycosis compared to free drug dispersion. These results agree with several studies, confirming the advantages of nanomedicine in treating ocular diseases [44,45,46]. Nano-delivery systems offer advantages in ocular disease therapy by lowering eye irritation, enhancing bioavailability by providing a route of entry to the eye, targeting the drugs to the right compartment of the eye, increasing residence time on the tear film, and enhancing corneal permeability. In these ways, nano-delivery systems overcome several barriers, including the muco-aqueous tear layer, the corneal epithelium, and the blood–retina barrier, which make the eye impermeable for most therapeutic agents [47].

## 4. Conclusions

The ability of the cationic nanocarriers (LP) to improve the corneal permeation and retention of the lipophilic drug STZN was successfully proven. The molar ratio of the lipid and the cationic SAA with its type dramatically affects the properties of the prepared LP. The optimized LP formula showed minute particle size, spherical morphology, and high entrapment efficiency. Additionally, the optimized LP formula maximized the ocular permeation and deposition of STZN, and the enhanced ocular contact time of the LP formula due to the mucoadhesion properties together with the small particle size assured intimate contact with the epithelial mucosal surface of the eye, preventing tear washout and consequently providing sustained drug release and a prolonged drug retention time. Furthermore, the in vivo studies revealed the superiority of the STZN-loaded LP in treating keratomycosis compared to the aqueous drug dispersion. Therefore, cationic nanocarriers (LP) have great potential to improve the corneal permeation and deposition of practically insoluble drugs.

## Figures and Tables

**Figure 1 pharmaceutics-14-02215-f001:**
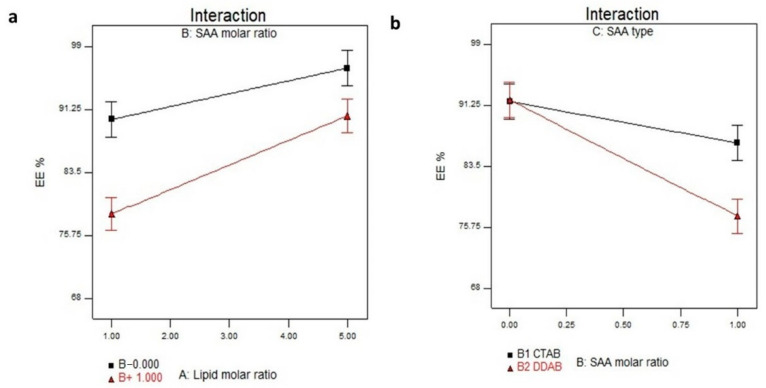
(**a**) Interaction plot for the effect of lipid molar ratio (X_1_) and cationic SAA molar ratio (X_2_) on EE%. (**b**) Interaction plot for the effect of and cationic SAA molar ratio (X_2_) and cationic SAA type (X_3_) on EE%. Abbreviation: SAA, surfactant; EE%, entrapment efficiency percent.

**Figure 2 pharmaceutics-14-02215-f002:**
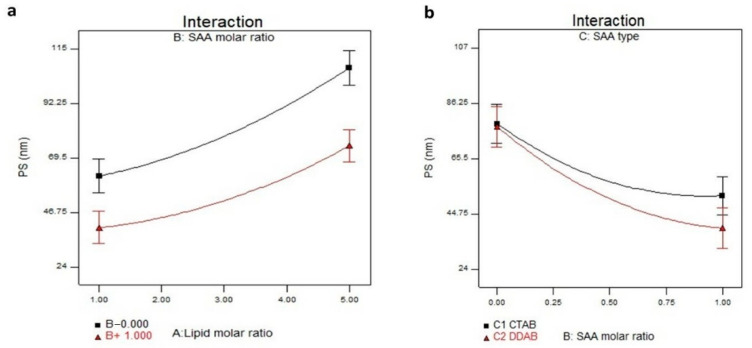
(**a**) Interaction plot for the effect of lipid molar ratio (X_1_) and cationic SAA molar ratio (X_2_) on PS. (**b**) Interaction plot for the effect of and cationic SAA molar ratio (X_2_) and cationic SAA type (X_3_) on PS. Abbreviation: SAA, surfactant; PS, particle size.

**Figure 3 pharmaceutics-14-02215-f003:**
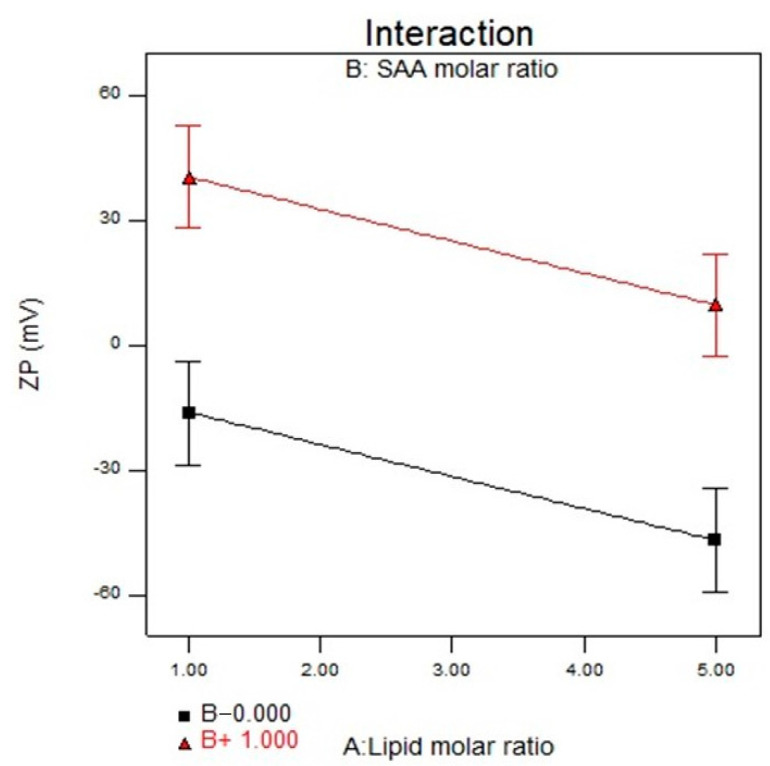
Interaction plot for the effect of lipid molar ratio (X_1_) and cationic SAA molar ratio (X_2_) on ZP. Abbreviation: SAA, surfactant; ZP, zetapotential.

**Figure 4 pharmaceutics-14-02215-f004:**
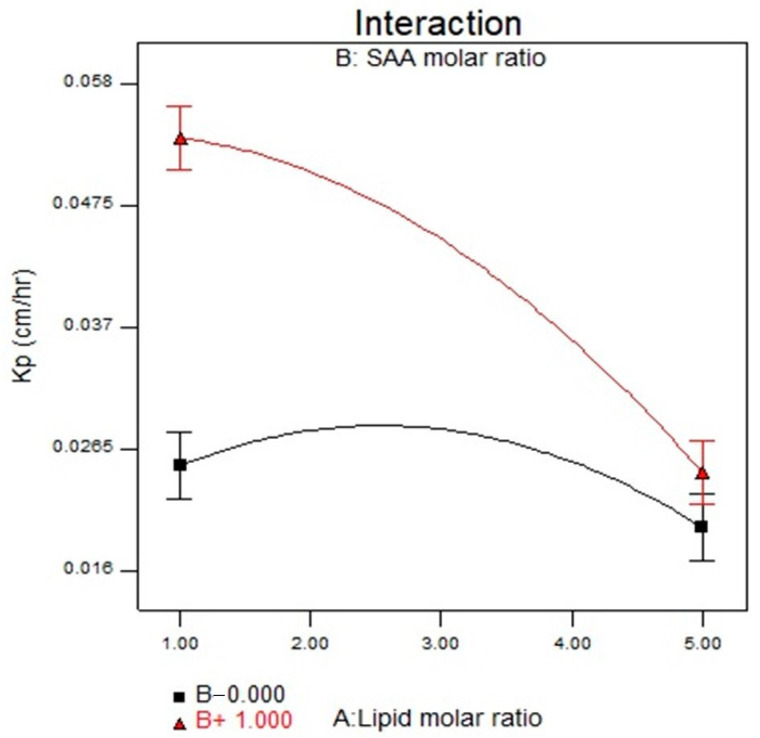
Interaction plot for the effect of lipid molar ratio (X1) and cationic SAA molar ratio (X2) on Kp. Abbreviation: SAA, surfactant; Kp, permeability coefficient.

**Figure 5 pharmaceutics-14-02215-f005:**
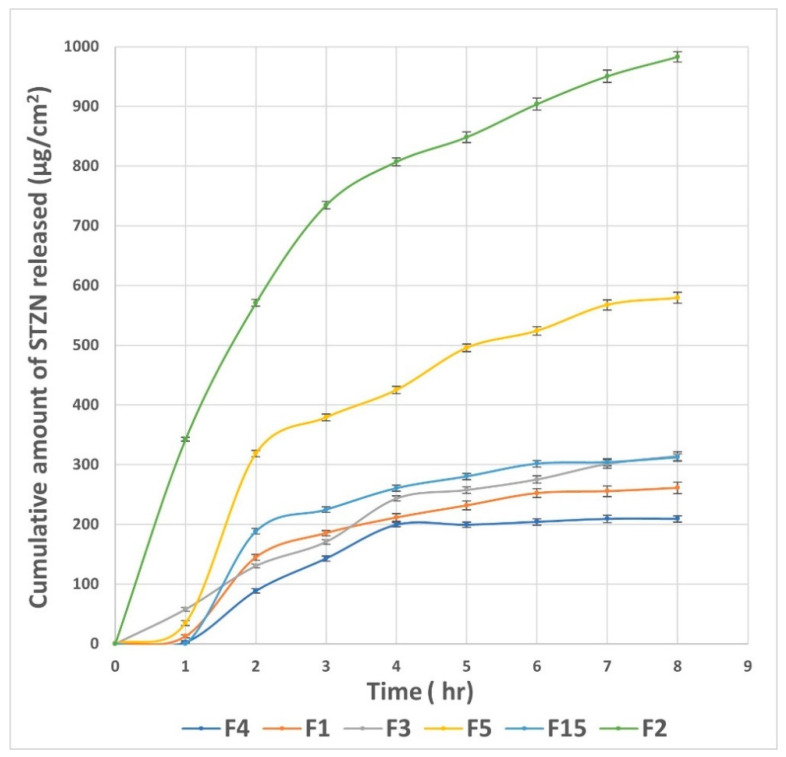
In vitro drug release profiles from different LP formulae. Abbreviation: LP, leciplex.

**Figure 6 pharmaceutics-14-02215-f006:**
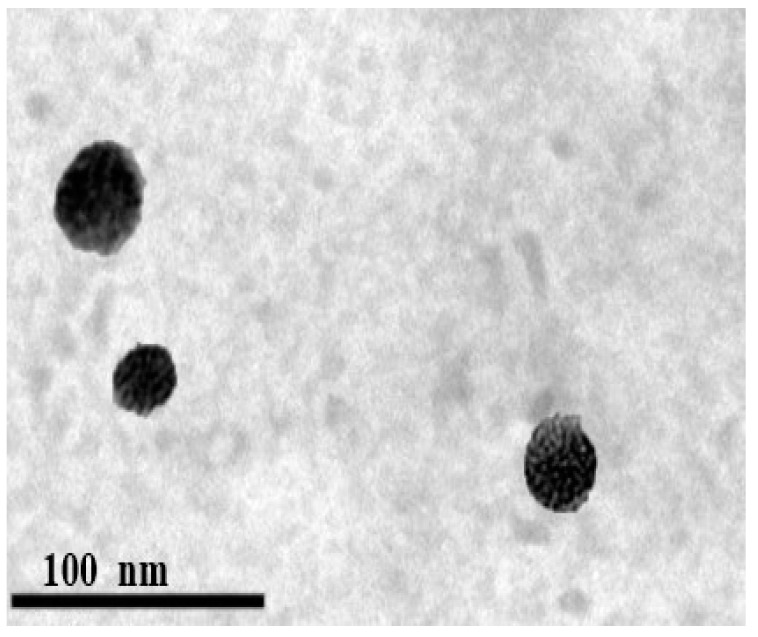
Morphology of the optimized LP. Abbreviation: LP, leciplex.

**Figure 7 pharmaceutics-14-02215-f007:**
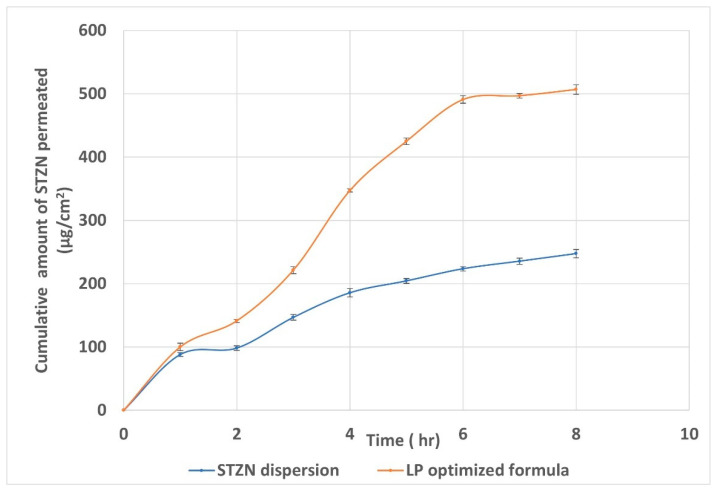
Cumulative amount of STZN permeated from the optimized LP formula and aqueous dispersion. Abbreviations: STZN, sertaconazole nitrate; LP, leciplex.

**Figure 8 pharmaceutics-14-02215-f008:**
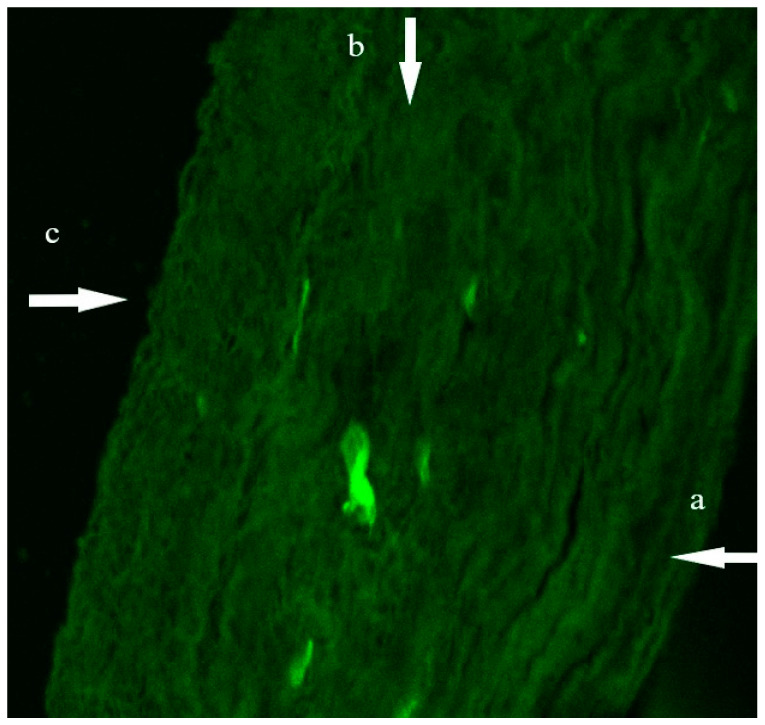
A tile scan confocal laser microscope photomicrograph of a longitudinal section in cow cornea treated with FDA-loaded LPs; a, epithelium; b, stroma; c, endothelium. Abbreviations: FDA, fluorescein diacetate; LPs, leciplex.

**Figure 9 pharmaceutics-14-02215-f009:**
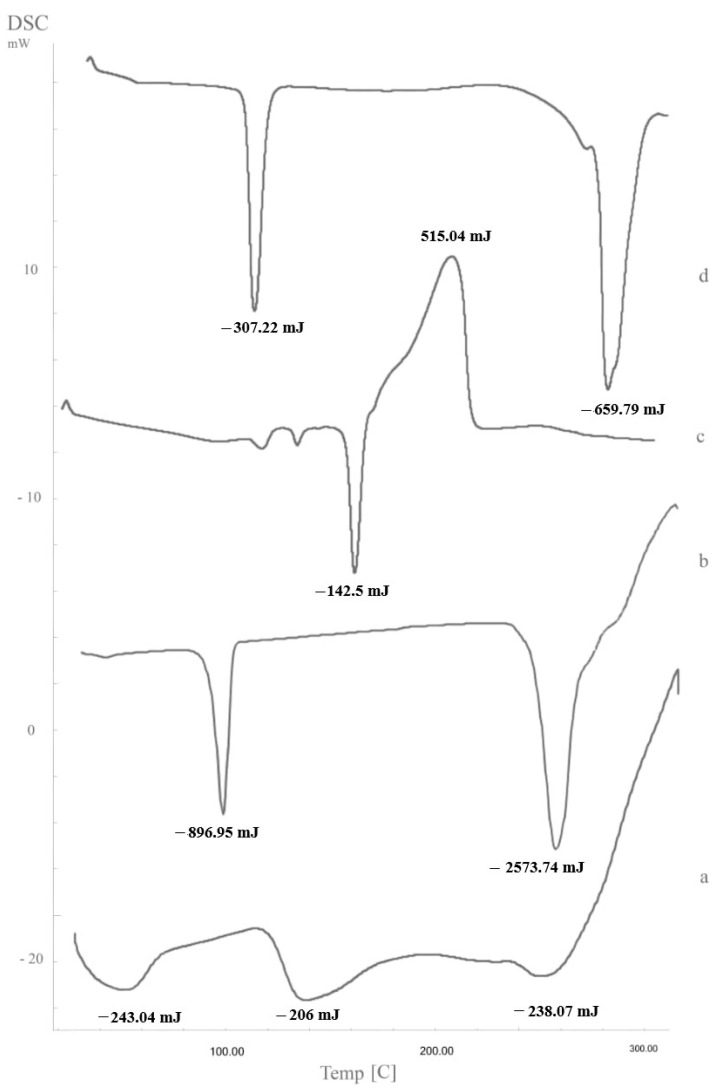
DSC thermograms; a, LP formula; b, CTAB; c, STZN; d, SPC. Abbreviations: DSC, differential scanning calorimetry; LP, leciplex; CTAB, cetyltrimethylammonium bromide; STZN, sertaconazole nitrate; SPC, soy phosphatidylcholine.

**Figure 10 pharmaceutics-14-02215-f010:**
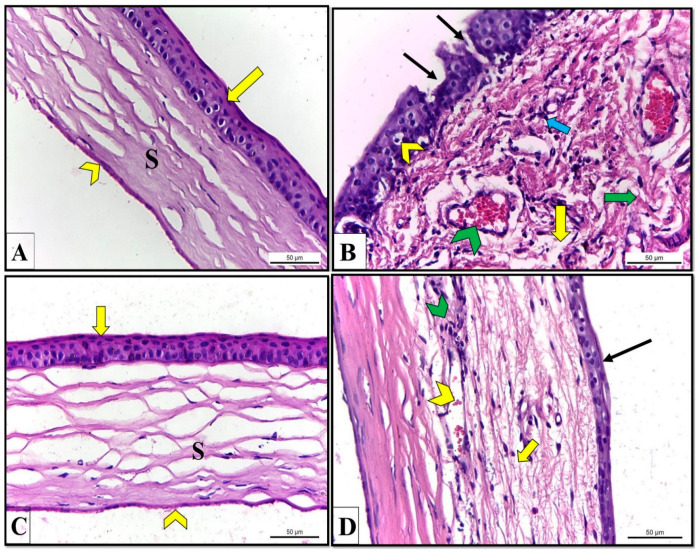
Corneal sections of albino rats. H&E stain X400. (**A**) Cornea of negative control rats (GPI) had normal stratified squamous epithelium (yellow arrow), stroma (S), and endothelium (chevron). (**B**) The cornea of rats with fungal keratitis (GPII) revealed increased corneal thickness, sloughing of epithelium portions (black arrows), cytoplasmic vacuolation of some epithelial cells (yellow chevron), dispersed collagen fibers (green arrow), a proliferation of inflammatory cells (blue arrow), and dilated blood capillaries engorged with blood (green chevron). (**C**) The cornea of rats treated with STZN-loaded LP (GP III) appeared to have the normal histological structure of stratified squamous epithelium (yellow arrow), stroma (S), and endothelium (yellow chevron). (**D**) corneal sections of rats treated with aqueous drug dispersion (GPIV) revealed partial recovery, including decreased corneal thickness, an epithelium with restored architecture but still without normal layer arrangement (black arrow), few inflammatory cells (green chevron), little collagen fibers dispersion (yellow arrow), and blood capillaries appeared narrower with little blood (yellow chevron).

**Table 1 pharmaceutics-14-02215-t001:** D-optimal design for optimization of STZN-loaded LP.

Factors (Independent Variables) for LP Design	Levels	
	Low	High
X_1_: lipid molar ratio	1	5
X_2_: SAA molar ratio	0	1
X_3_: SAA type	DDAB	CTAB
Responses (dependent variables)	Constraints	
Y_1_: EE (%)	Maximize	
Y_2_: PS (nm)	Minimize	
Y_3_: PDI	Minimize	
Y_4_: ZP (mV)	Maximize	
Y_5_: Kp (cm/h)	Maximize	

Abbreviations: SAA, surfactant; EE%, entrapment efficiency percent; CTAB, cetrimonium bromide; DDAB, dimethyldioctadecylammonium bromide; STZN, sertaconazole nitrate; PS, particle size; PDI, polydispersity index; SPC, soya phosphatidylcholine; ZP, zeta potential; Kp, permeability coefficient; and LP, leciplex.

**Table 2 pharmaceutics-14-02215-t002:** Experimental runs, independent variables, and measured response of the D-optimal experimental design of STZN-loaded LP.

Formulation Code	Lipid Molar Ratio (X_1_)	SAA Molar Ratio (X_2_)	SAA Type (X_3_)	EE % (Y_1_)	PS (nm) (Y_2_)	PDI (Y_3_)	ZP (mV) (Y_4_)	Kp (cm/h) (Y_5_)
F1	3	0	CTAB	90.52 ± 0.89	80.70 ± 0.21	0.248 ± 0.001	−32.5 ± 0.43	0.0228 ± 0.0001
F2	1	1	DDAB	68.50 ± 1.2	24.18 ± 0.11	0.235 ± 0.004	34.50 ± 0.69	0.04934 ± 0.0004
F3	5	1	DDAB	88.24 ± 0.92	59.11 ± 0.18	0.289 ± 0.009	6.49 ± 0.13	0.02261 ± 0.0002
F4	5	0	DDAB	96.02 ± 0.43	106.0 ± 0.34	0.226 ± 0.003	−38.70 ± 1.1	0.01991 ± 0.0003
F5	3	1	DDAB	81.21 ± 0.73	46.02 ± 0.17	0.213 ± 0.001	24.82 ± 0.21	0.04245 ± 0.0003
F6	1	0.5	CTAB	85.92 ± 1.67	40.10 ± 0.08	0.274 ± 0.005	32.60 ± 0.13	0.03644 ± 0.0001
F7	5	1	CTAB	92.69 ± 1.1	67.80 ± 0.29	0.238 ± 0.009	−28.13 ± 0.25	0.02577 ± 0.0003
F8	3	0.25	DDAB	94.71 ± 1.43	54.22 ± 0.11	0.288 ± 0.001	−29.90 ± 0.57	0.030584 ± 0.0004
F9	5	1	DDAB	87.22 ± 0.69	58.22 ± 0.32	0.289 ± 0.002	6.98 ± 0.11	0.02261 ± 0.0001
F10	5	1	CTAB	92.64 ± 1.1	82.59 ± 0.29	0.252 ± 0.006	27.81 ± 0.42	0.02577 ± 0.0003
F11	1	0.5	DDAB	84.60 ± 0.81	49.00 ± 0.23	0.266 ± 0.003	9.69 ± 0.12	0.02385 ± 0.0005
F12	1	1	CTAB	84.87 ± 1.71	39.70 ± 1.35	0.242 ± 0.006	54.60 ± 0.24	0.05770 ± 0.0001
F13	5	0	DDAB	95.89 ± 0.53	104.8 ± 0.41	0.226 ± 0.001	−38.10 ± 0.17	0.01992 ± 0.0002
F14	1	1	CTAB	83.12 ± 0.78	42.38 ± 0.34	0.252 ± 0.006	53.19 ± 0.18	0.05679 ± 0.0001
F15	1	0	DDAB	88.48 ± 0.98	61.00 ± 0.19	0.224 ± 0.004	−31.00 ± 0.24	0.02644 ± 0.0001
F16	1	0	CTAB	89.92 ± 0.71	63.56 ± 0.15	0.212 ± 0.008	−31.80 ± 0.09	0.025947 ± 0.0006
F17	5	0	CTAB	96.65 ± 0.80	106.84 ± 0.19	0.229 ± 0.002	−38.50 ± 0.62	0.01991 ± 0.0003
F18	5	0	CTAB	96.18 ± 0.59	106.11 ± 0.27	0.226 ± 0.001	−39.20 ± 0.41	0.01989 ± 0.0001
F19	3	0.5	CTAB	91.36 ± 1.3	56.26 ± 0.24	0.294 ± 0.003	−32.20 ± 0.31	0.02957 ± 0.0003

Abbreviations: SAA, surfactant; EE%, entrapment efficiency percentage; STZN, sertaconazole nitrate; PS, particle size; PDI, polydispersity index; Kp, permeability coefficient; LPs, leciplex; and ZP, zeta potential.

**Table 3 pharmaceutics-14-02215-t003:** Output data of the D-optimal designs analysis of LP formulations.

Source	EE (%)	PS (nm)	PDI	ZP (mV)	Kp (cm/h)
*p*-value	<0.0001	<0.0001	0.0624	<0.0001	<0.0001
Model	2FI	Quadratic	-	Linear	Quadratic
X_1_ = A = Lipid molar ratio	<0.0001	<0.0001	0.1184	0.0035	<0.0001
X_2_ = B = SAA molar ratio	<0.0001	<0.0001	0.0727	<0.0001	<0.0001
X_3_ = C = SAA type	0.0084	<0.0001	0.9096	0.7253	0.0899
Adequate precision R^2^	15.789 0.9040	17.652 0.9659	5.506 0.6944	11.559 0.7939	18.805 0.9663
Adjusted R^2^ (LP)	0.8559	0.9387	0.4499	0.7527	0.9393
Predicted R^2^ (LP)	0.7226	0.8641	−0.2181	0.6706	0.8660
Significant factors (LP)	X_1_, X_2_, X_3_	X_1_, X_2_, X_3_	-	X_1_, X_2_	X_1_, X_2_

Abbreviations: EE%, entrapment efficiency percentage; PS, particle size; PDI, polydispersity index; Kp, permeability coefficient; LP, leciplex; and ZP, zeta potential.

**Table 4 pharmaceutics-14-02215-t004:** Permeability parameters of STZN after application of STZN aqueous dispersion and LP.

Permeability Parameters	LP	STZN Aqueous Dispersion
The total amount of STZN permeated per unit area after 8 h (μg/cm^2^)	536.65 ± 1.95	247.38 ± 1.75
Apparent permeability coefficient (Kp) (cm/h)	0.0427 ± 0.0044	0.0153 ± 0.0076
Corneal deposition (μg/cm^2^)	348.859 ± 1.8	27.93 ± 0.53

Note: Data are presented as mean ± SD (n = 3). Abbreviations: STZN, sertaconazole nitrate; LP; leciplex; Kp, apparent corneal permeability coefficient.

**Table 5 pharmaceutics-14-02215-t005:** Assessment of serum inflammatory biomarkers and BDG of different groups.

	GP I (Negative Control)	GP II (Positive Control)	GP III	GP IV
hs-CRP	62.8 ± 2.1	227.8 ± 1.57	63.7 ± 0.61	112.7 ± 1.96
IL-23	7.8 ± 0.91	37.3 ± 1.18	6.4 ± 0.59	15.3 ± 1.09
BDG	3.4 ± 0.88	15.6 ± 1.2	3.3 ± 0.41	6.2 ± 0.72

Note: Data are presented as mean ± SD (n = 3). Abbreviations: hs-CRP, C-reactive protein; IL-23, interleukin-23; and BDG, beta-D-glucan.

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
