# Peer review of "Sertaconazole-Nitrate-Loaded Leciplex for Treating Keratomycosis: Optimization Using D-Optimal Design and In Vitro, Ex Vivo, and In Vivo Studies"

_pharmaceutics, 2022, doi:10.3390/pharmaceutics14102215_

Round 1

Reviewer 1 Report

The manuscript is a rigorous study of optimizing a cationic nanocarrier containing sertaconazole as an ocular drug formulation for treating keratomycosis. After the optimization of the formulation by using a design-of-experiment approach, the most optimal formulation was thoroughly characterized with appropriate in vitro, ex vivo and in vivo methods. The manuscript is clearly written and results are presented in a logical order taking the reader through the experiments and results nicely, while simultaneously building the proof of a successfully designed formulation. I have only a few minor suggestions for corrections and imrpovements:

1. On page 8, line 290: Correct the number for the standard deviation for 68.50. There is a full stop too much in the standard deviation value.

2. On page 11, first paragraph of text describing figure 5: Formulation 2 (F2) is not mentioned at all in the text, while it is still plotted in the figure. I assume this belongs to the series of F3 and F5 formulations and should be mentioned on line 382 and again on line 390.

3. It is very hard to see the green fluorescence in Figure 8. Would there be any better quality picture available for this?

4. In figure 9, unify in style and size the letters indicating different DSC thermograms. Also, use capital letters in the figure legend if letters in figure indicating thermorgrams are capital letters.

5.  The authors use s-genitive form on several places in the manuscript and I think this should be avoided in scientific texts, unless one refers to a person. For example, page 2, lines 51-52 it is written "...improves the active substance's solubility, permeability, and stability, enhancing the drug's ocular bioavailability." This should be written as "...improves the solubility, permeability, and stability of the active substance, enhancing the ocular bioavailability of the drug." All these s-genitive forms in the whole manuscript should be rewritten.

6. The conclusions section was a little bit thin in information. It could be improved by highlighting the most important results.

Author Response

Response to reviewer 1

Dear Reviewer

        We appreciate your efforts in reviewing the research article entitled (Sertaconazole nitrate-loaded leciplex for treating keratomycosis: optimization using D-optimal design and in-vitro, ex-vivo, and in-vivo studies). Modifications were made based on your reviewing results to improve the article and were highlighted in the revised manuscript.

Response to reviewer's comments

1-On page 8, line 290: Correct the number for the standard deviation for 68.50. There is a full stop too much in the standard deviation value.

Dear reviewer, thank you for your excellent revision; the SD value was correct. Please check Page 8, line 283

 2- On page 11, first paragraph of text describing figure 5: Formulation 2 (F2) is not mentioned at all in the text, while it is still plotted in the figure. I assume this belongs to the series of F3 and F5 formulations and should be mentioned on line 382 and again on line 390.

Dear reviewer, thank you for your excellent revision, F2 is mentioned now in the revised manuscript. Please check page 11, line 381.

3-It is very hard to see the green fluorescence in Figure 8. Would there be any better quality picture available for this?

Dear reviewer, thank you for your valuable comments: the figure was replaced with another one with a better resolution. Please check page 14, figure 8.

4- In figure 9, unify in style and size the letters indicating different DSC thermograms. Also, use capital letters in the figure legend if letters in figure indicating thermorgrams are capital letters.

Dear reviewer, thank you for your valuable comments: the figure was revised, and the style and the size of the letters were adjusted; also, small letters were used to comply with the ones used in the figure legend. Please check page 15, figure 9.

5-The authors use s-genitive form on several places in the manuscript and I think this should be avoided in scientific texts, unless one refers to a person. For example, page 2, lines 51-52 it is written "...improves the active substance's solubility, permeability, and stability, enhancing the drug's ocular bioavailability." This should be written as "...improves the solubility, permeability, and stability of the active substance, enhancing the ocular bioavailability of the drug." All these s-genitive forms in the whole manuscript should be rewritten.

Dear reviewer, thank you for your valuable comments, all the statements that contain s-genitive were revised and highlighted in the revised manuscript.

6-The conclusions section was a little bit thin in information. It could be improved by highlighting the most important results.

Dear reviewer, thank you for your valuable comments. The conclusion section was rewritten; please check page 17, lines 353-365.

Reviewer 2 Report

Accept with minor revision

1. Kindly recheck to the error bar in Figure 5, look like all error bar same

2. Figure 6 not justified, there is lot of nose on background

3. Figure 7 has also same error as figure 5. Please add the SD in error bar not 1% SD

4. Provide resolution in fig 8

5. Fig 9, provide the detail what is enthalpy value and add description impact on enthalpy. Provide the y bar

Author Response

Response to reviewer 2

Dear Reviewer

        We appreciate your efforts in reviewing the research article entitled (Sertaconazole nitrate-loaded leciplex for treating keratomycosis: optimization using D-optimal design and in-vitro, ex-vivo, and in-vivo studies). Modifications were made based on your reviewing results to improve the article and were highlighted in the revised manuscript.

Response to reviewer's comments

1-Kindly recheck to the error bar in Figure 5, look like all error bar same

Dear reviewer, thank you for your valuable comment; the SD values were added to figure 5. Please check Page 11, figure 5.

 2-  Figure 6 not justified, there is lot of nose on background

Dear reviewer, thank you for your valuable comment; the TEM figure was replaced with another one with less noise. Please check page 12, figure 6.

3-Figure 7 has also same error as figure 5. Please add the SD in error bar not 1% SD

Dear reviewer, thank you for your valuable comment; the SD values were added to figure 7. Please check Page 13, figure 7.

4-Provide resolution in fig 8

Dear reviewer, thank you for your valuable comments: the figure was replaced with another one with a better resolution. Please check page 14, figure 8.

5-Fig 9, provide the detail what is enthalpy value and add description impact on enthalpy. Provide the y bar

Dear reviewer, thank you for your valuable comments; the y-bar and the values were added to the figure; please check page 19, figure 9. The values were also mentioned, and the impact of these values was discussed. Please check  page 14, lines 457-467

Round 2

Reviewer 2 Report

Accepted after following correction.

1.  Figure 5, indicated that time is varies not cumulative release, kindly apply vertical SD bar not horizontal 

2. Figure 5, kindly star with zero. Add more justification why F2 have higher cumulative released in comparison to to other. 

3. figure 6 still confusing, what is other than black particles ? if possible focus on black particles and also write down resolution power 

4. Same mistake at figure 7. Change the SD bar apply in horizontal, Request to do modification carefully

5. Figure 8 just green color, write down the different part name. 

6. DSC figure need more clear, its look like stretch image, please write sentence very carefully. Line number 466 saying the drug in lipid bilayer and then after line mention that drug amorphous state . 

7. Why melting point of drug shift in leciplex 

Author Response

Response to reviewer 2

Dear Reviewer

        We appreciate your efforts in reviewing the research article entitled (Sertaconazole nitrate-loaded leciplex for treating keratomycosis: optimization using D-optimal design and in-vitro, ex-vivo, and in-vivo studies). Modifications were made based on your reviewing results to improve the article and were highlighted in the revised manuscript.

Response to reviewer's comments

1-Figure 5, indicated that time is varies not cumulative release, kindly apply vertical SD bar not horizontal .

Dear reviewer, thank you for your excellent revision; the SD value was corrected. Please check Page 11, Figure 5.

 2- Figure 5, kindly star with zero. Add more justification why F2 have higher cumulative released in comparison to to other. 

Dear reviewer, thank you for your excellent revision; the figure was revised. Please check Page 11, Figure 5.A justification was add, please check page 11, lines 390-393.

 3- figure 6 still confusing, what is other than black particles ? if possible focus on black particles and also write down resolution power        

Dear reviewer, thank you for your valuable comments: the figure was replaced with another one with a better resolution (300 dpi). Please check page 12, figure 6.

4-  Same mistake at figure 7. Change the SD bar apply in horizontal, Request to do modification carefully

Dear reviewer, thank you for your excellent revision; the SD value was corrected. Please check Page 13 figure 7

  1. Figure 8 just green color, write down the different part name. 

Dear reviewer, thank you for your valuable comments, the name of the different corneal layers were mentioned in the figure 7 and written in the figure caption; please check  figure 8 page 14.

  1.  DSC figure need more clear, its look like stretch image, please write sentence very carefully. Line number 466 saying the drug in lipid bilayer and then after line mention that drug amorphous state . 

Dear reviewer, thank you for your valuable comments. DSC figure is replaced with a more clear and more accurate version; please check page 15 figure 9, the DSC discussion was rewritten, and this phrase was omitted; please check page 14, lines 469-472.

7-Why melting point of drug shift in leciplex 

Dear reviewer, thank you for your valuable comments. the DSC discussion was rewritten, please check page 14, lines 469-472.